# Serological Number for Characterization of Circulating Antibodies

**DOI:** 10.3390/ijms20030604

**Published:** 2019-01-30

**Authors:** Andrea Palermo, Alexander Nesterov-Mueller

**Affiliations:** Institute of Microstructure Technology, Karlsruhe Institute of Technology (KIT), 76344 Eggenstein-Leopoldshafen, Germany; andrea.palermo@posteo.de

**Keywords:** antibodies, binding motif, humoral memory, vaccination

## Abstract

The dissociation constant of the circulating IgG antibodies is suggested to be proportional to the partial concentrations of these antibodies in blood serum in equilibrium. This coefficient, called serological number, is a dimensionless parameter and may be equal for all antibodies in a serum. Based on the serological number, we derived the equilibrium equation of the humoral immune system which allows estimating the number of different binding motifs in a serum. This equation also allows estimating the number of binding motifs of posttranslational and conformational nature. The feasibility of measuring the serological number via peptide arrays was demonstrated. Fifteen peptides with unique binding motifs were incubated and stained with the blood serum of a healthy adult at different dilutions. From these experiments, the serological number was determined. The serological number may explain the pre-existing antibody response after vaccination.

## 1. Introduction

The easy extraction of circulating serum antibodies from blood samples and their ability to selectively bind to targets involved in immune defense make serum antibodies very attractive for studying the immune system response. However, in many cases, the interpretation is complex and requires considering the entire antibody repertoire as, for example, in the case of influenza-specific antibody response after vaccination, which is shaped by pre-exposure history [1]. A straightforward approach to performing serum-antibody profiling is serum sample incubation with random peptide microarrays. Hereby, the reactivity of antibodies is measured using labeled secondary antibodies that bind to the constant region of the subset of serum antibodies [2]. In the case of a limited number of peptides (<1000), Greiff et al. have proposed to use multivariate regression analysis to predict the peptide binding signals of the serum antibodies [3]. Although this method does not take into account the information on amino acid position, the authors reported a successful prediction for up to 40%–50% of the binding profiles for highly diverse random antibody mixtures which are not dominated by a few antibodies. Legutki et al., claiming the limits of the state-of-the-art peptide technology, have proposed to use a chip with the same 10 M random peptides to detect the so-called serological immunosignatures—correlated patterns of the peptide binders specific for different infectious diseases and cancers [4]. Sticking to the same sequences was determined by the very high costs of masks used by the authors for lithographic peptide synthesis that makes it impossible, for example, to obtain the binding motifs, i.e., invariant amino acids, without fabricating new peptides.

Due to the large progress in high throughput sequencing, a recombinant method has been used to try and identify serum IgG binding targets [5,6]. Christiansen et al. used a bioinformatics approach in order to identify binding peptide motifs of interest based on clustering and contrasting to control samples from a phage display [7]. However, comparison of patient and control samples confirmed a major issue in phage display, namely, the selection of unspecific peptides and the necessity of their validation, for instance, with peptide arrays. Weber et al. have combined the screening via phage display with a substitutional analysis of the phage display hits via peptide arrays [8]. Using such an approach, 42 unique binding motifs for IgG antibodies were identified in the serum of a healthy donor. Such peptides include invariant combinations of amino acids and represent the epitopes of the circulating antibodies.

Searching binding motifs of serum antibodies is believed to give insight into systemic reactions of the humoral immune system because the invariant sequences may enable an easy identification of the involved proteins by searching modern databases. On the other side, such characteristics as the dissociation constant *K_D_*, or the fractional concentration *N*_0_, of antibodies seem to be less relevant for the systemic analysis without knowing an immunogen. In addition, both parameters cannot be obtained in a high throughput manner in comparison with the binding motifs. However, it is interesting that both parameters have the same unit “mol/L” [9] and can build the dimensionless serological number, or *S*-number:*S* = *K_D_*/*N*_0_(1)

Dimensionless combinations of parameters are frequently used to distinguish principal features of complex phenomena as, for example, the Reynolds number for characterization of the fluid flow. It is worth noticing that the *S*-number could be an intrinsic parameter of the immune system and may represent an optimum between the number of circulating antibodies and their affinity to the immunogens. The reason for this optimum may be the minimization of the high-affinity antibody concentration to get more space for antibodies against other pathogens. In the case of antibodies, one speaks about specific binding motifs, which can be obtained by so-called substitutional analysis (Figure 1) [8]: all amino acids in a peptide MVPEFSGSFPMR are substituted in each position against the 20 amino acids, while the rest of the sequence remains conserved, to define the invariant amino acids—namely, the binding motif. The binding motifs can be clearly expressed as X–A/I/P/V–P–E–F–X–G–A/S–X–P–X–X in Figure 1, or be heterogenic, allowing several invariant amino acids in a certain position in the sequence (as indicated by a slash). In following these considerations, we refer to the dissociation constant of an antibody *K_D_* and such binding motifs. In this paper, we investigate the potential of the *S*-number as an additional parameter for studying the immune response and demonstrate the possibility of its determination via peptide arrays.

## 2. Results

### 2.1. Analytical Estimation of the S-Number of Different Binding Motifs

The sum of the *S*-numbers over all subset antibodies of a serum sample gives the equation of equilibrium for the humoral immune system:(2)$$\overset{Z_{bm}}{\underset{k=1}{\sum }}S_{k}=\overset{Z_{bm}}{\underset{k=1}{\sum }}\frac{K_{Dk}}{N_{0k}}$$
Here, *Z_bm_* is the number of different binding motifs, *S_k_* is the serological number of the antibody with a number *k*, and *k* is the index of addition over all circulating antibodies. It is worth noticing that no special assumptions are needed to perform such additions. If the *S*-number relates to the entire humoral system, i.e., *S* = *S*_1_ = *S_k_*, Formula (2) can be simplified to
(3)$$S=\frac{\sum_{k=1}^{Z_{bm}}K_{Dk}}{\sum_{k=1}^{Z_{bm}}N_{0k}}.$$
For this transformation, the following formula was used: $\sum_{k=1}^{Z}\frac{a_{k}}{b_{k}}=N\frac{\sum_{k=1}^{Z}a_{k}}{\sum_{k=1}^{Z}b_{k}}$, if the ratio *a*_k_/*b*_k_ is constant for all *k*. After the addition over the fractional concentrations of all antibodies and averaging of the dissociation constants over Z*_bm_*, we obtain Formula (4), which states the relation between the *S*-number and the number of different binding motifs Z*_bm_*:(4)$$S=\;\frac{Z_{bm}\cdot \bar{K_{D}}}{N_{tot}},$$
whereby *N_tot_* is the total concentration of the antibodies in the serum. Applying in (4) the average $\bar{K_{D}}$ = 10^−8^ mol/L and *N_tot_* = 10 g/L for the IgG antibodies, as well *Z_bm_* = 42 reported for the serum used in this paper, *S* = 0.007 is obtained.

### 2.2. Dilution Measurements of the S-Number

Table 1 includes motifs that have been associated with antigens, like the polio vaccine or *Staphylococcus aureus* motifs, which the immune system of the healthy donor has faced in the past, as well as heterogeneous motifs with a large number of invariant amino acids. The heterogeneous motifs cannot be unambiguously associated with a specific pathogen. The corresponding *S*-numbers, measured according to (7), are presented in Table 1. The average value of the *S*-number over all 15 binding motifs is 0.0146 with a standard deviation of 0.0095.

## 3. Discussion

Equation (4), which is based on the of the *S*-number approach, gives the value range of the *S*-number from 0.007 for the 42 binding motifs to 0.049 for 300 binding motifs. It is interesting that the *S*-numbers, which are measured independently according to (7), fall in this range. If we put the average *S*-number, 0.0146, for the serum sample measured in this paper into (4), we will obtain 90 different specific binders. We note that only 42 specific binder motifs were found by Weber et al. [8] for this serum sample using phage display technology. This discrepancy may have two reasons. First, the phage library did not include the rest of the linear specific binders. Secondly, the missing specific binders may have a conformational or posttranslational nature, which cannot be registered either with phage display or with linear peptide arrays. It is worth noticing that the derivation of (4) does not include any assumptions about the epitope nature and, thus, can be used to estimate the number of nonlinear epitopes that are very difficult to determine with any other methods.

In Table 1, we observe an increase of the *S*-number from 0.003 to 0.044, which still remains in the estimated *S*-number interval, but cannot be explained even by relatively large measurement errors. This deviation can be explained by differences in the IgG antibody affinities to the peptides integrated in pathogenic proteins and the peptide spots synthesized on a solid support. For instance, the 15th peptide exhibits an *S*-number of ~0.044, which is a factor ~3 higher than the averaged *S*-number. This may mean that this binding motif embedded in a protein exhibits a *K_D_* smaller by a factor ~3, i.e., has better affinity to the pathogen than the single peptide on a solid support.

The *S*-number can be explained on the cellular level when considering the mechanisms of the long-term humoral memory (Figure 2).

As previously described, the limited number of survival niches in the bone marrow requires the regular elimination of long-lived plasma cells for “old” antigens by generating plasmablasts with a new specificity [10]. The number of plasmablasts which transform to long-term plasma cells in the bone marrow is equal to the number of the “old” plasma cells that are substituted in survival niches and then emitted into the blood. It was estimated that the humoral memory for an “old” antigen would wane at a frequency of 0.1% for each generation of new plasmablasts [11]. This mechanism keeps a constant level of circulating antibodies for antigens which have been already recognized by the adaptive immune system. Using (4), the *S*-number for “old” antibodies supplied to the blood due to the cellular exchange in the bone marrow has the form
(5)$$S\;=\;Z_{\mathrm{bm}}\;\bar{K_{D}}/\Delta N\cdot \alpha \;=\;\beta K_{\mathrm{Dc}}/(\Delta N*\alpha )\;=\;S_{c}\beta /\alpha ,$$
where α is the number of antibodies generated by a single plasma cell that has left the bone marrow, β is the number of antibodies on a single B cell with the improved affinity, and *S_c_* = *K_Dc_*/Δ*N*. The cellular *S*-number *Sc* characterizes the generation of new plasma cells with a fractional concentration ΔN from the B cells with an improved cellular dissociation constant *K_Dc_* towards the antigen. The generated plasma cells in the bone marrow will, in turn, build a fractional *S*-number *S*_k_ = *K*_Dc_/Δ*N* = *K_Dk_*/*N*_0k_. Here, the equations *K_Dc_* = β⋅*K_Dk_* and *N*_0k_ = β⋅Δ*N* were taken into account. Thus, we see that the bone marrow transfers the serologic proportions from the B cells, with the improved affinity, to the circulating antibodies. The reason for a larger fractional concentration of the plasmablasts and their corresponding precursors—memory cells, with a smaller affinity to the antigen—may be due to the fact that B cells with smaller affinities reenter additional rounds of mutative replication and, therefore, they have a higher probability to be developed in larger amounts at the plasmablast stage.

The *S*-number approach allows developing a systemic view on the generation of serum antibodies by vaccinations. As mentioned in the introduction, the influenza-specific antibody response after vaccination is shaped by pre-exposure history [1,12,13,14]. Recent studies have shown that the vaccination against influenza can boost up to 60% of pre-existing antibodies [15,16]. If numerous low-affinity plasmablasts for a new viral antigen contact the survival niches, they eliminate 0.1% of the previously stored plasma cells. According to the *S*-number, the number of low-affinity memory cells, and correspondingly long-lived plasma cells in the bone marrow niches, is larger than the number of plasma cells for high-affinity antigens. Thus, significantly more low-affinity pre-existing antibodies than high-affinity pre-existing antibodies will be generated, as is observed after vaccination.

## 4. Materials and Methods

### 4.1. Equation for Measurement of the S-Number in Array Format

Equation (2) describes the dependence of the dissociation constant between a protein and a ligand in equilibrium, which can be measured optically by labeling analyte molecules:(6)$$I_{obs}=I_{sat}\frac{N}{K_{D}+N},$$
where *I_sat_* is the maximum fluorescent signal at saturation, *N* the concentration of the analyte (antibody in our case) in the solution, and *K_D_* is the equilibrium dissociation constant [17]. *K_D_* can be measured in parallel for many molecules if one of the binding partners is arrayed in spots on a solid support and the other carries a fluorescent label, and if its concentration, *N*, is known [18].

Using the definition of the *S*-number (1) and introducing the dilution factor γ = *N/N*_0_, Formula (6) can be transformed to
(7)$$I_{obs}=I_{sat}\frac{1}{S/\gamma +1\;}.$$

Obviously, the condition *S* = γ is reached at *I_obs_* = *I_sat_*/2. This means that if the patient’s serum will be constantly diluted, the dilution coefficient will be equal to the *S*-number at the signal *I_obs_* = *I_sat_*/2. Actually, the *S*-number can be measured by other *I_obs_* and dilution values. The general formula has the form *S* = α∙k, by *I_obs_* = *I_sat_*/(α + 1), where α is an arbitrary positive number and can be defined from the fluorescent signal saturation curve. We chose *I_obs_* = *I_sat_*/2, because this value refers to IC50, known as half maximal inhibitory concentration and can be easily measured by the available analytic techniques. Please note, to determine the *S*-number, it is not necessary to know either the concentration of antibodies or coefficients that converts the fluorescent signal to concentration units. Such a simple approach is especially useful for measuring *S*-numbers by studying a large number of antibody interactions with combinatorically synthesized peptide arrays.

### 4.2. Methods to Measure the S-number

Blood serum of a healthy individual was used. Ethical clearance was obtained from the state chamber of physicians of Baden-Wuerttemberg (reference number: F-2011-044 and F-2011-044#A1). Identical peptide arrays were used for dilution experiments. The peptide content included binding motifs for IgG serum antibodies of the healthy individual that were already determined by Weber et al. [8]. Every peptide was represented by 10 to 40 spots. The serum was serially diluted, ranging from 1:800 to 1:12.5, and incubated overnight. The serum was diluted in PBS-T containing 10% (v/v) Rockland blocking buffer (Rockland Immunochemicals Inc., Pottstown, PA, USA). After the incubation, the arrays were stained with the secondary antibody (goat) anti-human IgG (Fc γ) AF647 (Jackson Immunoresearch, West Grove, PA, USA) at 1 µg/mL. Fluorescent signals were measured with an InnoScan scanner (Innopsis, Carbonne, France). The collected fluorescent signals from the secondary antibodies (Appendix A) and the dilution values were fit using Formula (6) to calculate the *S*-numbers for each binding motif (Appendix A). The absolute intensity for each pixel within a spot and the spot duplicate was averaged and background corrected. Therefore, the average pixel intensity of a control area, not containing any peptide spots, was subtracted from the calculated spot intensity. Fits with a coefficient of determination *rs* < 0.9 were discarded.

## 5. Conclusions

The *S*-number, as an additional parameter for the characterization of circulating antibodies, was presented. It includes both the information about the antibody affinity and the fractional concentration.

Based on the assumption of a constant *S*-number for an antibody ensemble, the formula was derived that combines the total concentration of antibodies, their average dissociation constant, and the number of different binding motifs.

The determination of the *S*-number was experimentally demonstrated. Therefore, the arrays with the same peptide content representing the binding motifs were incubated with the serum samples of a single human donor at different dilutions and stained with secondary anti-human IgG antibodies. The signals obtained were approximated with the kinetic equation and, consequently, the antibody *S*-numbers for each binding motif were determined.

The experimentally measured *S*-numbers revealed values of the same order of magnitude for all motifs, and were approximated with the formula derived for the antibody ensemble with a constant *S*-number. These results should not be considered as evidence for an equal *S*-number for all IgG antibodies in human serum, and serve solely as an example of the measurement of the *S*-number via peptide arrays. In addition, the nature of the *S*-number is discussed from the point of view of B-cell ontogeny. The *S*-number may explain the generation of pre-existing antibodies after vaccination against influenza.

## Figures and Tables

**Figure 1 ijms-20-00604-f001:**
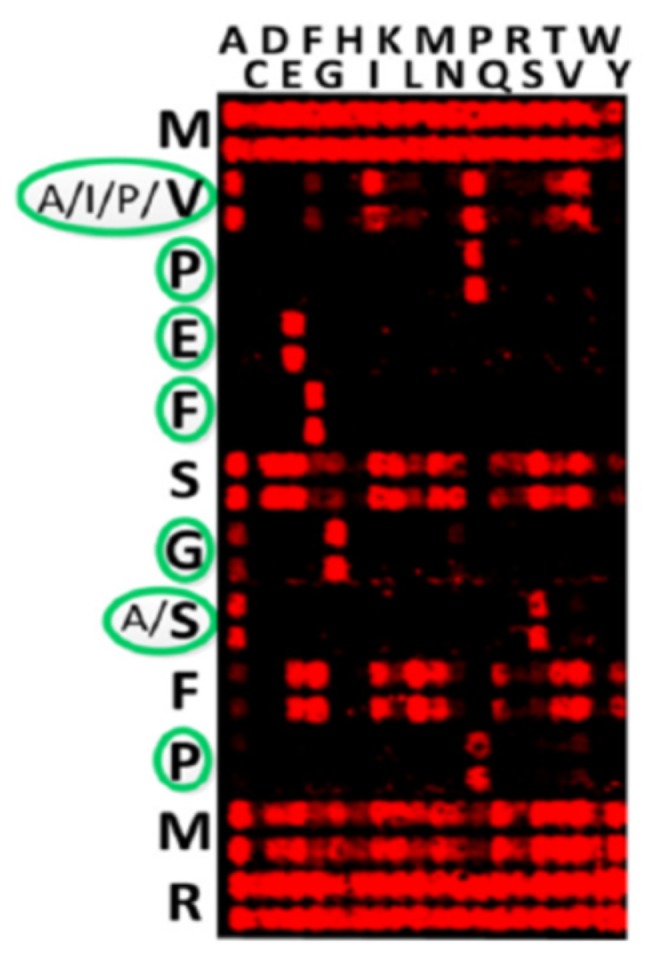
Substitutional analysis of peptide MVPEFSGSFPMR reveals the binding motif X–A/I/P/V–P–E–F–X–G–A/S–X–P–X–X [8]. The double spot array was used. Reproduced with permission, Copyright 2017, ELSEVIER.

**Figure 2 ijms-20-00604-f002:**
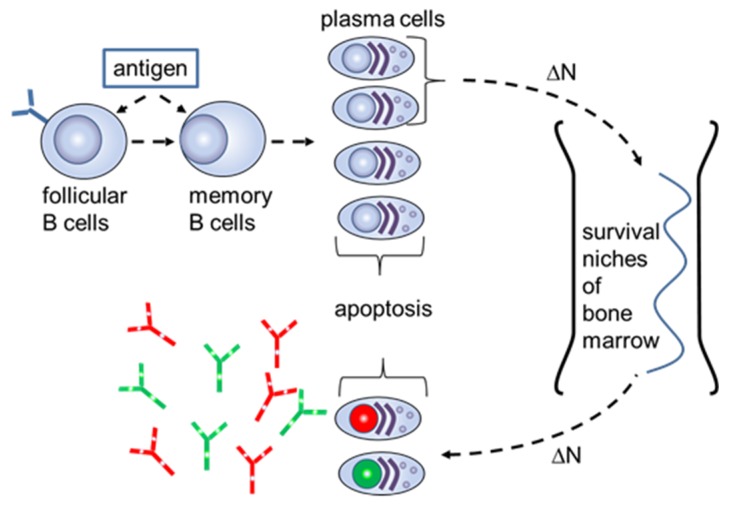
Schematic of the long-term humoral memory via elimination of the long-lived plasma cells for “old” antigens (red and green nucleus) with plasmablasts with a new specificity (lilac nucleus).

**Table 1 ijms-20-00604-t001:** Peptides corresponding to the binding motifs and their measured *S*-numbers.

N	Amino Acid Sequence	Characteristics of the Motif	*S*-number	Standard Deviation in %
1	GGQVRSIHSGPT	heterogeneous motif	0.00267	37
2	KEVPALTAVETGAT	LXAXETX motif group, poliovirus motif	0.00386	31
3	MVPEFSGSFPMR	*Staphylococcus aureus* motif	0.00620	56
4	LIADLNAESTSR	heterogeneous motif	0.00775	22
5	VLSSTAIKVDSV	heterogeneous motif	0.00865	49
6	VMSVNASTTAAN	heterogeneous motif	0.01183	41
7	QMKAWFPQTTYD	KXXFPQXT motif	0.01219	48
8	LRPNAVQTDTLA	heterogeneous motif	0.01302	39
9	SWVLTATETGSS	LXAXETX motif group, poliovirus motif	0.01427	34
10	NPVEDYLDYSVI	NPVEXXX motif	0.01490	60
11	ETKSDDMLLSNV	heterogeneous motif	0.01537	31
12	AKIRMFLDTDYK	heterogeneous motif	0.01841	58
13	VDTINLPQNTIQ	heterogeneous motif	0.02059	49
14	TALDAVSTGFSW	heterogeneous motif	0.02597	42
15	QHWPTNVDSVTV	heterogeneous motif	0.04362	24

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
