# Peer review of "Serological Number for Characterization of Circulating Antibodies"

_ijms, 2019, doi:10.3390/ijms20030604_

Round 1
Reviewer 1 Report
The authors have addressed all my comments and I believe the revised manuscript is much improved
Author Response
The authors have addressed all my comments and I believe the revised manuscript is much improved.
English language and style are fine/minor spell check required.
An English native speaker (Canadian with a PhD degree) has conducted spell check.
Reviewer 2 Report
Authors has addressed major concerns.
There is another major issue: the conclusion is missing for the article. It would be really helpful to wrap up outcomes from study in conclusion section.
Author Response
English language and style are fine/minor spell check required
An English native speaker (Canadian with a PhD degree) has conducted spell check.
Authors have addressed major concerns. There is another major issue: the conclusion is missing for the article. It would be really helpful to wrap up outcomes from study in conclusion section.
We have introduced the conclusion section to describe outcomes from study.
Reviewer 3 Report
The authors have made some changes according to previous reviews, but the big issue of using a single human sample has not been addressed. This is needed to make the manuscript rise to the level of IJMS. I, therefore, recommend rejecting the manuscript in its current form.
Author Response
The authors have made some changes according to previous reviews, but the big issue of using a single human sample has not been addressed. This is needed to make the manuscript rise to the level of IJMS. I, therefore, recommend rejecting the manuscript in its current form.
The S-number formulas and the method to measure the S-number via peptide arrays are the main results of our study. We introduce into the manuscript the conclusion section to more precisely summarise our results and to address the issue of using a single human sample. We ask the reviewer to take into account these important revisions.
Proving the constant value of the S-number for other/all IgGs or serum samples was not the goal of this paper. Would the experiments with a second (third?) serum give the evidence of the constant S-number for all humans? To do this, several independent medical trails are necessary, which are beyond of our competence and the scope of the Journal. For example, we use the donor’s binding motifs, which were previously obtained in other studies (cited in the manuscript) via advanced biological methods as phage display.
In our paper, we present to the scientific community the theory and the experimental method to get more information from the blood serum or other liquid human samples. The concept of S-number is not focused on IgG antibodies. It could work for some subclasses, for example, neutralizing antibodies or even other protein families that participate in self-organizing processes. We believe the S-number and the methodology to measure it will be of interest for the broad readership and stimulate new interdisciplinary research projects.
In addition, the science knows many cases of publishing pure formulas, which were proved only several decades after publishing. The most pronounced example is E=mc2. Did the publication of this formula in 1905 have no scientific value until the supporting experimental results were published by the chemists Otto Hahn in 1930is?
With best wishes
Authors
This manuscript is a resubmission of an earlier submission. The following is a list of the peer review reports and author responses from that submission.
Round 1
Reviewer 1 Report
In this article, the authors propose to use a new parameter, S, that correlated the Kd of a given antibody and its fractional concentration. They propose that this S value averaged among all antibodies is a constant among individuals, and, therefore, by knowing the the average Kd and the total antibody concentrations, assuming a constant S value, one can estimate the number of different binding motifs within the serum without doing complicated capture and affinity assays.
This is an interesting concept, and the authors present a reasonable rationale why the S number would be a constant. However, to determine if the equation holds and that the S number is indeed constant, they need to be able to use the equation predictively, not just retrospectively with a single serum sample. Effectively the authors use a known serum sample to determine the S number and then show that the S number holds for that serum sample. This argument is circular and proves nothing. Besides this, the authors should show some of the raw data used for calculation of the S numbers they calculate for individual antibodies in table 1.
Reviewer 2 Report
The phrase in the discussion section: "It has been known for many years that the influenza-specific antibody 146 response after vaccination is shaped by pre-exposure history" better fits to the introduction. I would recommend authors to rephrase it.
Materials and Methods section requires more details.
It is not clear whether or not the Figure 1 is produced by authors. If so, why is it located in Introduction? As the similar peptide analysis was conducted in study, it worth including corresponding diagrams.
Reviewer 3 Report
Palermo et al develops a model to predict number of binding motifs in a serum using the serological number (S-number). Overall, this manuscript is not well written and very hard to read. The authors should also spend more effort explaining the S-number with some applications. The results are not very convincible to me merely based on data from one individual.
1) The authors should at least add another individual to show the reproducibility of this method.
2) How well S-number is correlated with affinity or binding strength?
3) What are binding motives (line 12) or binding motifs?
4) Concertation -> Concentration (line 53)
5) S-number should be written out at the first place. The terminology use should be consistent, such as S-number, s number and S number …
6) KD -> KD (line 70)